# Regulating Enzymatic Antioxidants, Biochemical and Physiological Properties of Tomato under Cold Stress: A Crucial Role of Ethylene



Yousry Bayoumi [1], Sabah Osman [1], Abdelwahab Etman [1], El-Samahy El-Semellawy [2], Svein Ø. Solberg [3,*] and Hassan El-Ramady [4,*]

1  Physiology & Breeding of Horticultural Crops Laboratory, Horticulture Department, Faculty of Agriculture, Kafrelsheikh University, Kafr El-Sheikh 33516, Egypt
2  Vegetable Research Department, Horticulture Research Institute, Agriculture Research Center, Giza 12619, Egypt
3  Faculty of Applied Ecology, Agriculture and Biotechnology, Inland Norway University of Applied Sciences, 2318 Hamar, Norway
4  Soil and Water Department, Faculty of Agriculture, Kafrelsheikh University, Kafr El-Sheikh 33516, Egypt
*  Correspondence: svein.solberg@inn.no (S.Ø.S.); hassan.elramady@agr.kfs.edu.eg (H.E.-R.)

**Abstract:** The production of tomato under low-temperature stress in the open fields is a challenge faced by many farmers. The current study compares the use of different ethylene treatments to accelerate the fruit ripening of tomato during two successive seasons under cold stress. The treatments included foliar application of ethrel (2500, 5000, and 7500 ppm) in the open field at the mature green stage, dipping collected fruits in ethrel solution (1000, 1500, and 2000 ppm) right after harvest, and application of gaseous ethylene (100, 200, and 300 ppm) to the harvested fruits. The effects were compared to untreated fruits (control). Characteristics, such as physical properties (ripening, weight loss, firmness, decay, and fruit color), chemical properties (ascorbic acid, acidity, total soluble sugars, and pigments), and enzymatic activities (polygalacturonase and pectin methylesterase), were sampled throughout the storage period. In general, the ethylene gas application was the most effective method in accelerating the fruit ripening process compared to the other methods applied. The highest vitamin C total soluble solid contents and redness parameters were found after applying the highest dose of ethylene gas (300 ppm). This indicates that the ripening of tomato fruits, which are cultivated under cold stress conditions as found during the early summer season in a Mediterranean climate, might be harvested at a mature green stage and exposed to ethylene application.

**Keywords:** weight loss; firmness; decay; color index; ascorbic acid; acidity; pigments; enzyme activity; ripening

## 1. Introduction

Tomato (*Solanum lycopersicum* L.) is one of the most popular vegetables in the world and is used fresh or processed [1,2]. Tomato has several health benefits, especially with regard to the development of chronic degenerative diseases [3]. Tomato fruit is identified as a "poor man's orange". These fruits are rich in vitamins (e.g., A and C) and antioxidants (e.g., lycopene, and β-carotene), which can aid the prevention of contracting illness (e.g., cancer and heart disease). Tomatoes are also known to prevent and deactivate free radicals or reactive oxygen species, and can act as an effective eliminator of superoxide, hydrogen peroxide, singlet oxygen, and other free radicals [4,5]. Therefore, it is recommended to consume about 100 g of tomato fruits daily to improve the immune system, lower cholesterol, and reduce blood pressure [6].

The maturation process of the tomato fruit goes from green to mature green fruits when a climacteric process starts with the production of low amounts of ethylene before

it regularly increases until fruits ripen [7]. The ripening is the last stage of the fruit development, involving biochemical and physiological changes controlling the flavor, color, aroma, and texture of fruits [8]. The process is regulated by ethylene, a gaseous plant hormone. Tomato cultivation in cool climates (or during the Mediterranean winter and early summer) causes cold stress, which slows down the degradation of chlorophyll and synthesis of lycopene. This challenges the production. However, many growers appreciate this period for a potentially higher income if they succeeded in producing high-quality tomatoes [9,10]. Thus, tomato growers in northern Egypt tend to harvest tomatoes already cultivated in December and January in open field at the mature green stage, then accelerating their ripening artificially using ethylene to sell their production at a high price. The mean temperature (min. and max.) in Egypt ranged from 12.5 to 24.5 °C during the study (January to May), causing a weak and slow growth in the tomatoes. Applying ethylene is the best solution under these conditions [11]. The most common ripening agents are ethylene gas, ethrel, acetylene, ethylene glycol, ethanol, methanol, propylene, methyl jasmonate, and rosmarinic acid [12].

Ethylene gas ($C_2H_2$) is a natural plant hormone produced by climacteric plants to stimulate fruit ripening. The artificial use of ethylene is a method that has been used commercially in several countries [7]. Much is known about the stimulating effect of ethylene on the ripening process, but for farmers' practical aspects, such as doses, application time and mode of application are important. Factors, such as concentration, exposure time, temperature, relative humidity, and application methods, are of importance [7]. Ethrel (also called ethephon, $C_2H_6ClO_3P$) is considered an ethylene precursor and an ethylene-releasing compound [13]. Ethrel has been registered as a plant regulator at the US Environmental Protection Agency (EPA) since 1973 and can promote fruit ripening and reduce postharvest losses [14]. The application of ethrel on tomato fruits is known to accelerate the color change (from green to red) and affect physicochemical properties, such as total soluble solids (TSS) and acidity. The increase in TSS during fruit ripening may come as a result of the hydrolysis of starch and other polysaccharides to soluble sugar forms. Different types of anabolic and catabolic processes take place in the fruits as part of the ripening [4]. Ali et al. [15] reported that tomato fruits treated with ethylene, in contrast to untreated tomato fruits, exhibited maximal postharvest quality in terms of biochemical composition, antioxidant activity, and carotenoid and flavonoid contents.

Several studies are available on the effect of ethylene on fruit ripening, but our study will discuss three different application methods with different concentrations under cold stress as a main target. To achieve this main aim, various physical, chemical, and enzymatic characters were investigated as response variables. The current study will also focus on the role of ethylene in accelerating tomato fruit ripening and improving the quality under cold stress (i.e., early summer season) as found in a Mediterranean climate in northern Egypt.

## 2. Materials and Methods

### 2.1. Planting Material and Study Location

The cultivation was carried out in two successive early summer seasons (2020 and 2021) on a private farm at El-Khawaled Elbalad village (31°16′20.82″ N and 30°46′50.68″ E), Sidi Salem district, Kafr El-Sheikh governorate, Egypt. Tomato seedlings of 'Super Strain B' (F1 hybrid, from the Nunhems Netherlands BV Company, Nunhem, the Netherlands) were used in both seasons. 'Super Strain B' is the most common variety in Egypt for early summer season (under cold stress). Seedlings were obtained from the nursery at Faculty of Agriculture, Kafrelsheikh University, Egypt. The seedlings were transplanted in January in both seasons (the mean temperature was 7–21 °C). Irrigation, fertilization, and other horticultural practices were performed according to the recommendations from the Egyptian Ministry of Agriculture and Land Reclamation. Surface irrigation was applied with 8–12-day intervals, depending on the weather and the need, ending up with around 12 irrigations over a season. Harvesting was performed at a mature green stage (about 90 days from transplanting) and the tomato fruits were hand-picked in early morning

and thereafter transferred to the Horticulture Department Lab of the Sakha Horticultural Research Station for further treatments and storage.

*2.2. Applied Treatments*

Fruit ripening treatments were divided into three groups. The first group was foliar application of ethrel in the field at the mature green fruit stage, four days before harvesting. Three doses were examined (FE at 2500, 5000, and 7500 ppm) with fruits harvested 4 days after spraying and transferred to the Lab of Horticulture Department at the Sakha Horticultural Research Station. The second group was dipping the newly harvested tomato fruits in an ethrel solution for five minutes. Three doses were examined (DE at 1000, 1500, and 2000 ppm). The third group was ethylene gas exposure of the newly harvested tomato fruits. Three doses were applied (GE at 100, 200, and 300 ppm). A control with untreated fruits was included (Figures 1 and 2). The treatment codes were as follows: FE1, FE2, FE3, DE1, DE2, DE3, GE1, GE2, GE3, and the untreated fruits (control). The fruits were stored for five weeks at room temperature at the Horticulture Research Station, Sakha, Kafr El-Sheikh (Figure 3).

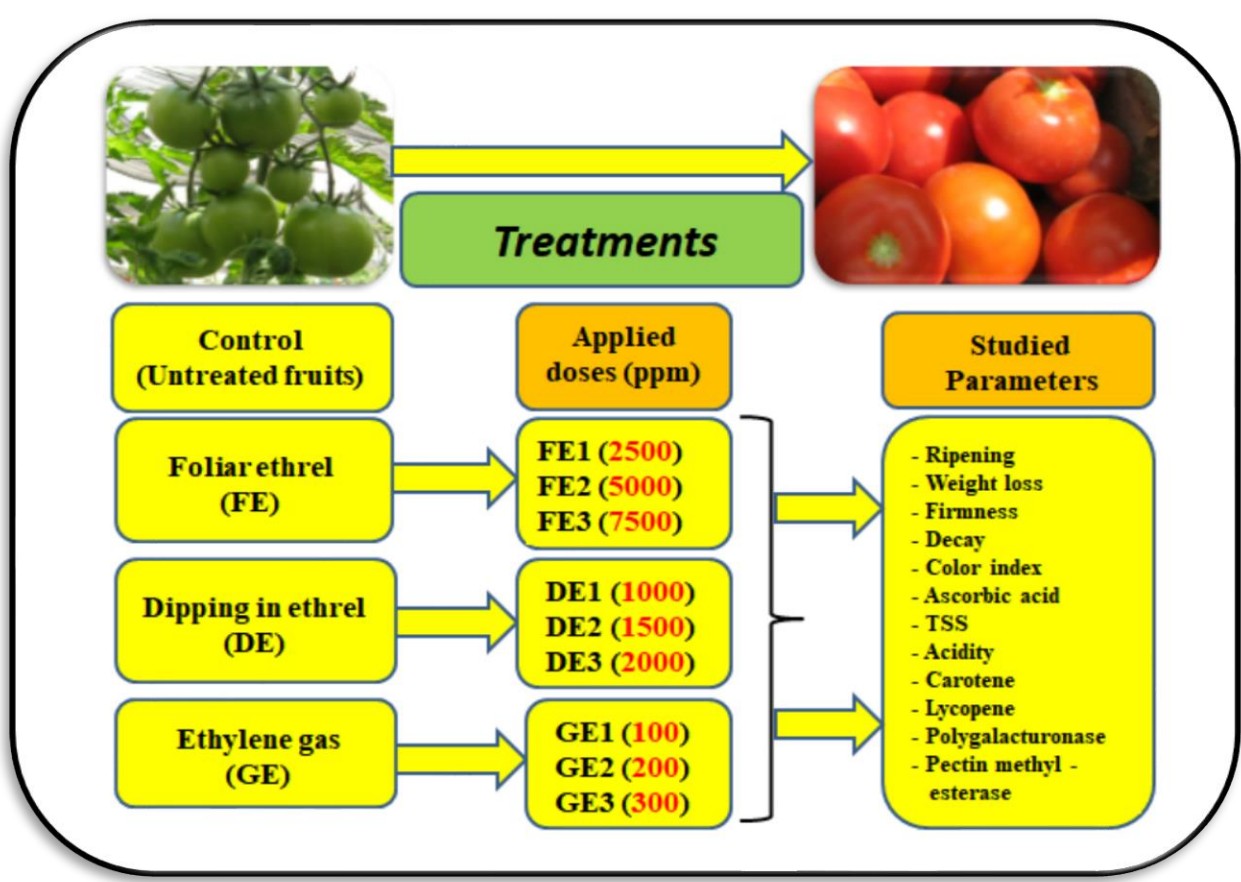

**Figure 1.** Overview of the applied treatments in the current study with applied doses (ppm) and the studied parameters.

Ethrel or Ethylene gas used in the experiments was obtained from ethephon compound (2-Chloroethyl phosphonic acid 48% SL). Ethrel compound was imported from CAM company for Agrochemicals (Ferty Agro Company, Singapore), the common name is Ethephon H, and its use is as a plant growth regulator from the group of ethylene generators. Ethylene gas was released from ethrel, which was put into airtight plastic boxes. The fruits and a cup containing 15 mL of 40% sodium hydroxide (NaOH) were put in the boxes. Calculated amounts of ethrel that would release ethylene gas of 100, 200, and 300 ppm were added quickly to the cup of NaOH, and the boxes were immediately closed. All boxes were

opened after 24 h. All tomato fruits selected for quality measurements were uniform in shape, color, and size (with an approximate diameter of 6–7 cm). Treated tomato fruits were then air dried, packed in foam plates with a capacity of 10 fruits per plate, and covered by plastic film with small holes for aeration and directly kept at room temperature to monitor physical and chemical fruit properties. Three replicates were used, and each treatment contained 30 fruits in both seasons. The sampling time for the different measurements were performed according to Table 1.

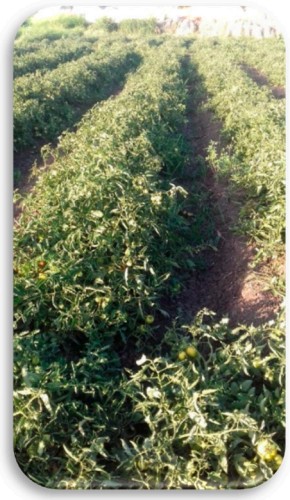 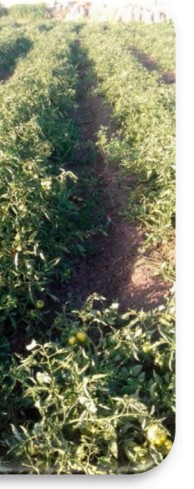 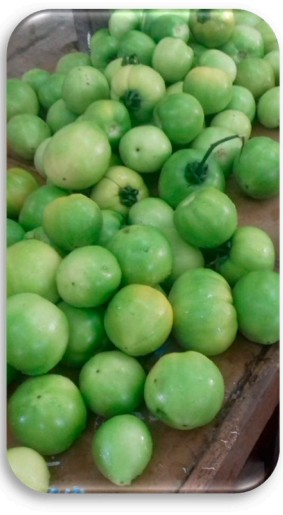 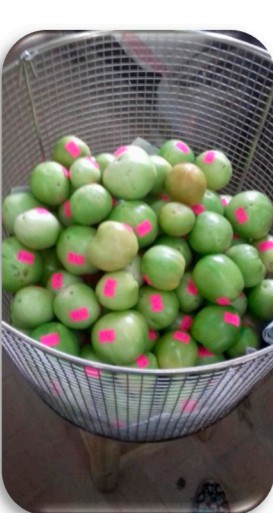

Growing tomato in the field      Harvesting tomato fruits at mature green stage

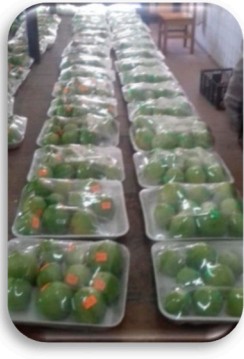 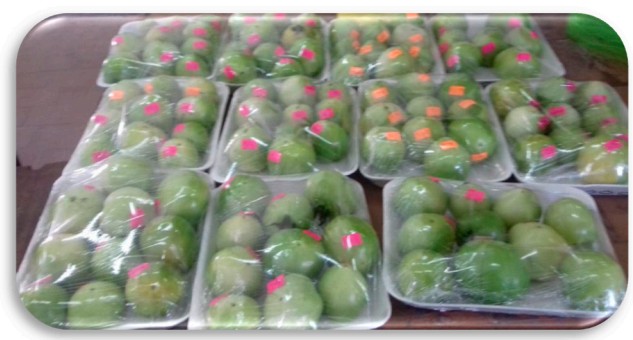

Treatment tomato fruits using different doses of ethrel and ethylene and then packing for storage

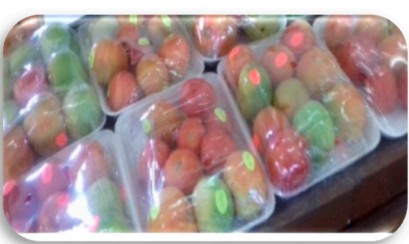 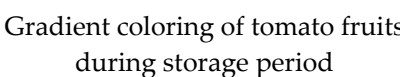 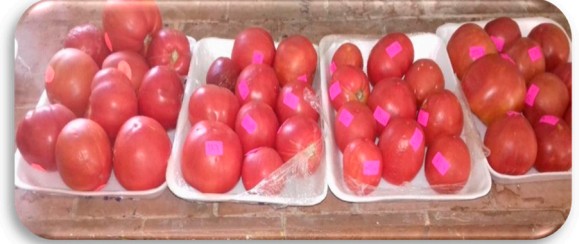

Gradient coloring of tomato fruits during storage period

Tomato fruits at ripening stage

**Figure 2.** General overview of the steps of the experiment from field to harvesting the fruits, packing the fruits after applying the treatments, and ripening of the fruits during storage.

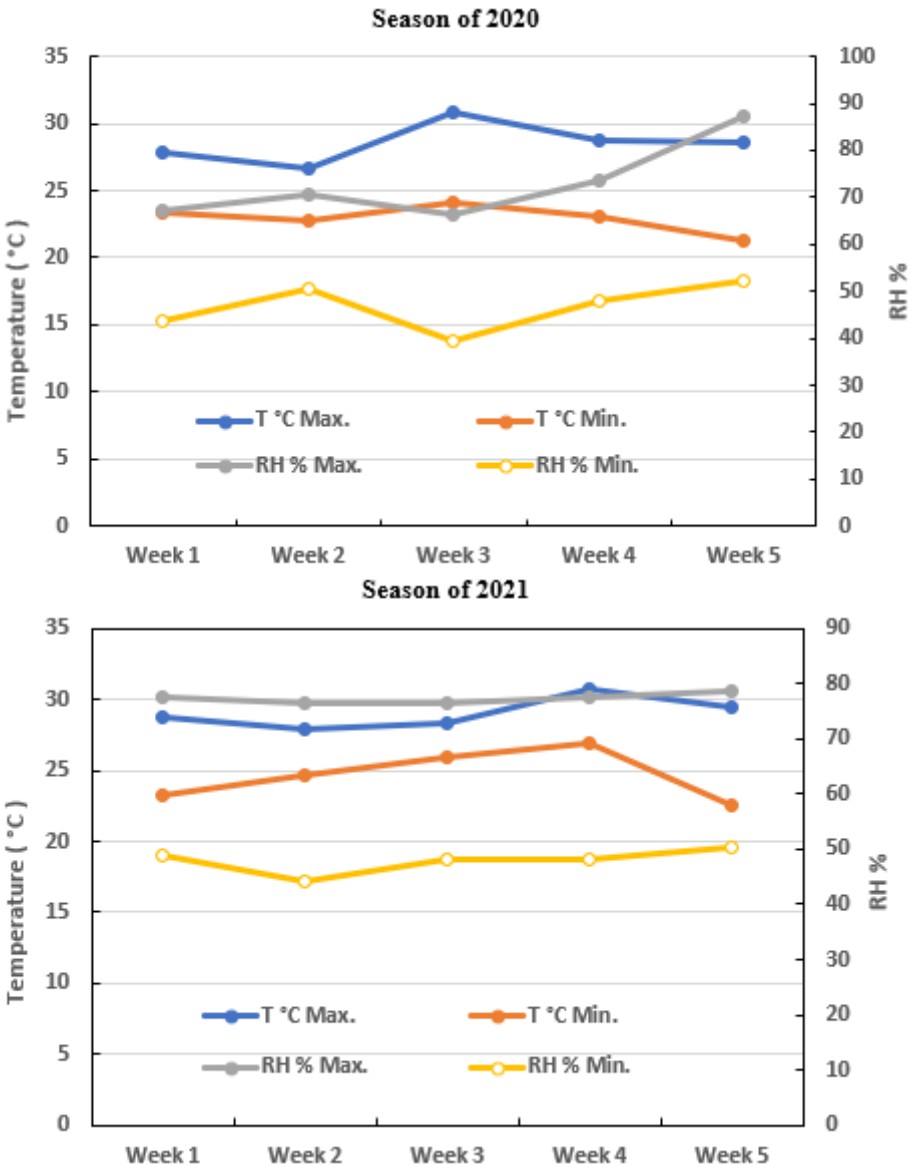

**Figure 3.** Average maximum and minimum room temperature and relative humidity during the storage periods of the tomato fruits in both seasons.

**Table 1.** Overview of the measured parameters and the sampling time.

| Measured Parameters | Sampling Time (in Days after Treatment/Putting into Storage) |
|---|---|
| *Physical parameters of fruits* | |
| Ripening (%) | At day 4, 8, and 12 |
| Weight loss, firmness | At day 8, 16, 24, and 32 |
| Decay (%) | At day 24, 28, and 32 |
| Color index (redness, lightness, and yellowness) | At day 10 and 30 |
| *Chemical parameters of fruits* | |
| Ascorbic acid, TSS, and acidity | At day 8, 16, 24, and 32 |
| Pigments (total chlorophyll, carotene, and lycopene) | At day 28 |
| Enzymes activity (polygalacturonase and pectin methyl esterase) | At day 10 and 21 |

### 2.3. Physical Parameters

Days required for ripening: fruits were observed daily and recorded for their color, and the days were counted.

Weight loss (%): It was calculated by using the following formula:

$$\text{Total weight loss } (\%) = \frac{Initial\ weight - final\ weight}{Initial\ weight} \times 100 \tag{1}$$

Firmness (kg mm$^{-2}$): It was determined by Magnus pressure tester. The firmness of tomato fruits was measured using push–pull dynamometer (Models FD, and FDN, Italy). The dynamometer is calibrated for use in horizontal position. Reading was recorded after pushing and pulling into the fruits.

Decay (%): It was determined by visual observation. Un-marketable tomatoes including fruits with various spots developed on the peel, as well as rotten, decayed and shriveled fruits, were considered as rotten. Decay % was calculated as the ratio between the number of fruits infected by microbes or showing rotting and number of sample fruits initially stored.

Fruit color: Digital phenotyping technique was used for measuring color by using the tomato analyzer (TA) software application, called Color Test (TACT), according to Darrigues et al. [16]. Color index of the tomato fruits was determined by measuring the values of "a", "b", and "L" using the software application by TACT. Concerning the "a value", it indicates the ratio of red to green color, and "b value" indicates the ratio of yellow to blue color, whereas "L value" indicates the ratio of white to black color. Hue angle (H* = $\tan^{-1}(b*/a*)$) was calculated based on values of "a" and "b" parameters [17]. This tool allows for accurate quantification of color and color uniformity, and the tool allows the scanning devices to be calibrated using color standards with a colorimeter and from scanned images. Images of fruit were taken with a digital camera on a black background are also appropriate, and then collecting, importing, and analyzing color from images are available in the TACT manual.

*2.4. Chemical Parameters*

Ascorbic acid content: It was estimated by 2,6-Dichlorophenol-indophenol visual titration method, according to A. O. A. C. [18], by putting 100 g of the fruits on 100 mL of oxalic acid 6%, then it was mixed in a blender and filtered with gauze, then 10 mL of the juice was placed in a glass and supplemented to 100 mL of 3% oxalic acid. Then, 10 mL of the previous solution was taken, dripped onto it, and the volume of the dye used was calculated. Then, by means of the strength of the dye, the amount of vitamin C in the fruits were able to be calculated.

Total soluble solids: Total soluble solids (TSS) percent was recorded in fruit juice by using a hand refractometer method. A drop of tomato juice from the fruit pulp was placed on the prism of the refractometer and TSS was recorded as % from a direct reading of the apparatus (refractometer).

Total titratable acidity: Total titratable acidity (TTA) percent was determined of tomato juice as citric acid percentage by titration with 0.1 N sodium hydroxide, according to A. O. A. C. [18].

Fruit pigments: Chlorophyll (Chl.), carotene (Car), and lycopene (Lyc) in tomato fruits were analyzed by spectrophotometer, according to the method of Nagata and Yamashita [19]. They were extracted with acetone and hexane (4:6), then optical density of supernatant at 663 nm, 645 nm, 505 nm, and 453 nm were measured by spectrophotometer for total Chl., Car., and Lyc., respectively, and calculated by the following equations:

$$\text{Total Chlorophyll} = 0.999\ A_{664} - 0.0989\ A_{647}$$

$$\text{Carotene} = 0.216\ A_{663} - 1.22\ A_{645} - 0.304\ A_{505} + 0.452\ A_{453}$$

$$\text{Lycopene} = -\ 0.0458\ A_{663} + 0.204\ A_{645} + 0.372\ A_{505} - 0.0806\ A_{453}$$

Enzymes activity: Polygalacturonase (PGU) and pectin methyl esterase (PME) activities in tomato fruits were determined after 10 and 21 days from starting storage of

the treated fruits using polygalacturonic acid and citrus pectin, respectively, according to Li et al. [20].

**Polygalacturonase activity:** PGU activity was determined through quantifying the amount of reducing groups expressed as galacturonic acid units, liberated during the incubation of 1 mL of 1% (*w/v*) citrus pectin, prepared in 0.2 M phosphate buffer (pH 7.2) with 500 μL of the enzyme at 37 °C for 30 min, by di-nitro-salicylic acid (DNSA) method. One unit of polygalacturonase activity was defined as the amount of enzyme required to release 1 μmol of galacturonic acid per minute under standard assay conditions 20 and expressed as units per liter (U/L).

**Pectin methyl esterase activity:** PME activity was determined using 50–500 μL of tomato juice, 50 g $L^{-1}$ solution of apple pectin in distilled water, and a 0.1 g $L^{-1}$ solution of bromothymol blue in a 0.003 M phosphate buffer. The starting pH value of these solutions was adjusted to 7.5 with 2 M NaOH. The zero-absorbance value was first set using distilled water. The pectin and bromothymol blue solutions were then mixed in a thermos-stated (25 °C) cuvette, and the initial absorbance at 620 nm was determined. The reaction was then started by adding the juice, and an absorbance rate decrease was recorded. PME activity of 1 U was defined as an absorbance decrease of 0.1 per minute.

### 2.5. Statistical Analyses

Completely randomized design system was used in both seasons and the experimental design contained ten treatments (ethrel applications) with three replicates. All the obtained data were statistically examined by variance analysis, and means were compared by Duncan's multiple range tests, following Snedecor and Cochran [21]. All analyses were performed by "M-STAT" software version 5.4.

## 3. Results

### 3.1. Physical Parameters of Tomato Fruits

#### 3.1.1. Ripening Process

All tomatoes reached 100% ripening within 12 days from harvesting the fruits at a mature green stage, but there were significant differences between the treatments in how fast the 100% ripening stage was achieved. Regarding GE treatment, the shortest time was obtained with the highest dose (300 ppm ethylene). Around 8 days was required to reach 100% ripening with 300 ppm ethylene gas, followed by the medium dose (200 ppm) that reached around 90% maturation after 8 days. Additionally, the foliar application of ethrel at the highest dose (7500 ppm) resulted in 90% ripening after 8 days in one of the seasons and 80% in the other season, which is not significantly different from ethylene at the medium dose (Table 2). The untreated fruits (control) showed the slowest ripening. So, both ethylene gas and foliar ethrel applications hastened the ripening process by about 4 days on average compared to the control in both seasons (Table 2). Furthermore, the data showed that all ethylene treatments (all applied doses) or all foliar applications of ethrel (all applied doses) accelerated the fruit ripening compared to the dipping treatments and the control.

#### 3.1.2. Weight Loss (%)

There was a significant effect of the different ethrel and ethylene applications on the weight loss of the tomato fruits. The total weight loss was highest with the ethrel applications (Figure 4). Both foliar and dipping applications of ethrel resulted in higher weight losses than the control and the ethylene treatments. Foliar ethrel application at the highest dose (7500 ppm) recorded the highest weight loss, and this was found at all sampling times (8, 16, 24, and 32 days after harvesting). Overall, untreated tomato fruits (the control) showed the lowest weight loss but did not significantly differ from the ethylene gas treatments.

**Table 2.** Effects of ethrel and ethylene applications on the ripening % of the tomato fruits 4, 8, and 12 days after the treatment in 2020 and 2021.

| Treatments | Ripening (%) of Stored Tomato Fruits after Treatment | | | | | |
|---|---|---|---|---|---|---|
| | 4 Days | 8 Days | 12 Days | 4 Days | 8 Days | 12 Days |
| | | 2020 Season | | | 2021 Season | |
| Control | 0.00 e | 18.50 f | 100 a | 0.00 e | 14.8 g | 100 a |
| FE1 | 14.8 d | 37.0 e | 100 a | 18.5 d | 25.9 fg | 100 a |
| FE2 | 18.5 d | 55.6 d | 100 a | 25.9 cd | 48.2 e | 100 a |
| FE3 | 48.2 bc | 92.6 ab | 100 a | 48.2 b | 81.5 bc | 100 a |
| DE1 | 18.5 d | 37.0 e | 100 a | 25.9 cd | 40.5 ef | 100 a |
| DE2 | 37.0 c | 51.9 d | 100 a | 37.0 bc | 51.9 e | 100a |
| DE3 | 40.0 bc | 77.8 c | 100 a | 48.2 b | 55.6 de | 100 a |
| GE1 | 40.8 bc | 70.4 c | 100 a | 37.0 bc | 70.4 cd | 100 a |
| GE2 | 51.9 ab | 88.9 b | 100 a | 48.2 b | 92.6 ab | 100 a |
| GE3 | 63.0 a | 100.00 a | 100 a | 63.0 a | 100.00 a | 100 a |
| F-test | ** | ** | NS | ** | ** | NS |

Treatments (in ppm): foliar ethrel included FE1 (2500), FE2 (5000), and FE3 (7500); dipping ethrel included DE1 (1000), DE2 (1500), and DE3 (2000); and ethylene gases were GE1 (100), GE2 (200), and GE3 (300). NS means non-significant, whereas ** means significant at $p < 0.01$. Means followed by the same letter in same column are not significantly different at the 0.05 level, according to Duncan's multiple range test.

### 3.1.3. Fruit Firmness

The effects of ethrel and ethylene on tomato fruit firmness (kg mm$^{-2}$) are shown in Table 3. In general, fruit firmness decreased over time. The untreated fruits (control) and ethylene (all doses) recorded the highest firmness values at all sampling times. Concerning the ethrel treatments, foliar application showed lower firmness values than the dipping method.

**Table 3.** Effect of ethrel and ethylene application on the tomato fruit firmness (kg mm$^{-2}$) 8, 16, 24, and 32 days after the treatment in 2020 and 2021.

| Treatments | Firmness (kg mm$^{-2}$) of Stored Tomato Fruits after Treatment | | | | | | | |
|---|---|---|---|---|---|---|---|---|
| | 8 Days | 16 Days | 24 Days | 32 Days | 8 Days | 16 Days | 24 Days | 32 Days |
| | | 2020 Season | | | | 2021 Season | | |
| Control | 320 a | 300 a | 250 a | 225 a | 285 a | 242 a | 210 a | 190 a |
| FE1 | 295 a–d | 250 cd | 220 bc | 180 e | 230 c | 220 b | 175 bc | 150 cde |
| FE2 | 270 de | 240 de | 210 cd | 152 f | 220 c | 195 c | 170 bc | 135 fg |
| FE3 | 250 e | 230 e | 190 d | 145 f | 192 d | 185 d | 151 c | 130 g |
| DE1 | 290 bcd | 253 cd | 231 abc | 200 bcd | 240 bc | 225 b | 185 ab | 147 cde |
| DE2 | 273 cde | 253 cd | 230 abc | 195 cde | 240 bc | 225 b | 180 b | 145 def |
| DE3 | 270.0 de | 250 cd | 220 bc | 190 de | 235 bc | 220 b | 173 bc | 140 efg |
| GE1 | 310.0 ab | 275 b | 250 a | 215 ab | 285 a | 240 a | 200 ab | 165 b |
| GE2 | 303.3 ab | 268 bc | 240 ab | 210 abc | 250 b | 235 ab | 193 ab | 160 bc |
| GE3 | 300 abc | 265 bc | 235 ab | 200 bcd | 240 bc | 225 b | 190 ab | 155 bcd |
| F-test | ** | ** | ** | ** | ** | ** | ** | ** |

Treatments (in ppm): foliar ethrel included FE1 (2500), FE2 (5000), and FE3 (7500); dipping ethrel included DE1 (1000), DE2 (1500); and DE3 (2000), and ethylene gases were GE1 (100), GE2 (200), and GE3 (300). Symbol ** means significant at $p < 0.01$. Means followed by the same letter in same column are not significantly different at the 0.05 level, according to Duncan's multiple range test.

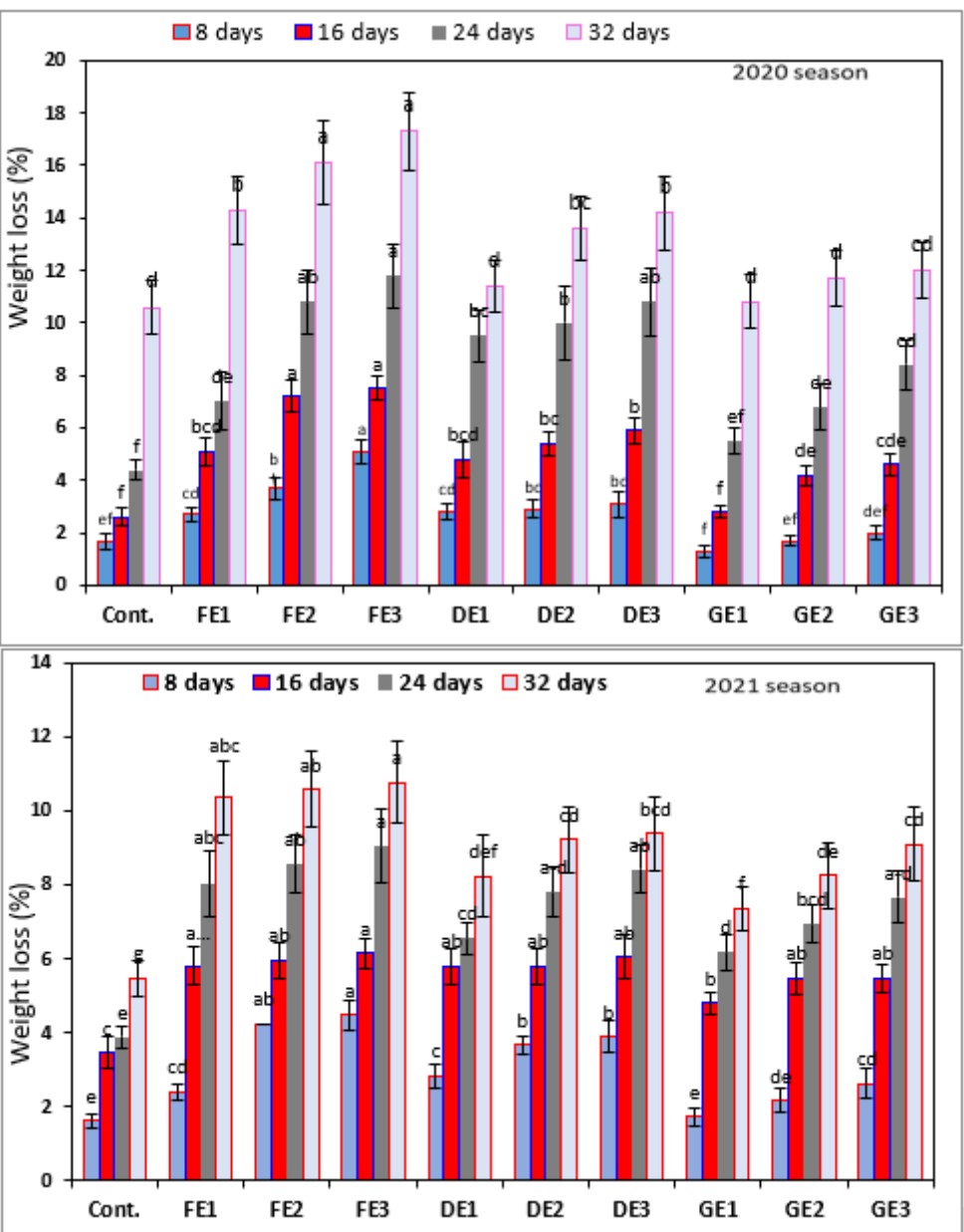

**Figure 4.** Effects of ethrel and ethylene application on weight loss (%) of the tomato fruits 8, 16, 24, and 32 days after the treatment in 2020 and 2021. Means followed by the same letter in same storage time are not significantly different at a 0.05 level, according to Duncan's multiple range test. Treatments (in ppm): foliar ethrel included FE1 (2500), FE2 (5000), and FE3 (7500); dipping ethrel included DE1 (1000), DE2 (1500), and DE3 (2000); and ethylene gases were GE1 (100), GE2 (200), and GE3 (300).

### 3.1.4. Decay (%)

In general, the decay rate of the tomato fruits increased over time. For all ethrel treatments the decay started to appear on day 24 in storage. Untreated fruits showed the lowest decay rate (Figure 5). The maximum decay was obtained with foliar ethrel application at the highest dose (7500 ppm). The ethylene treatments resulted in lower decay values than the ethrel treatments.

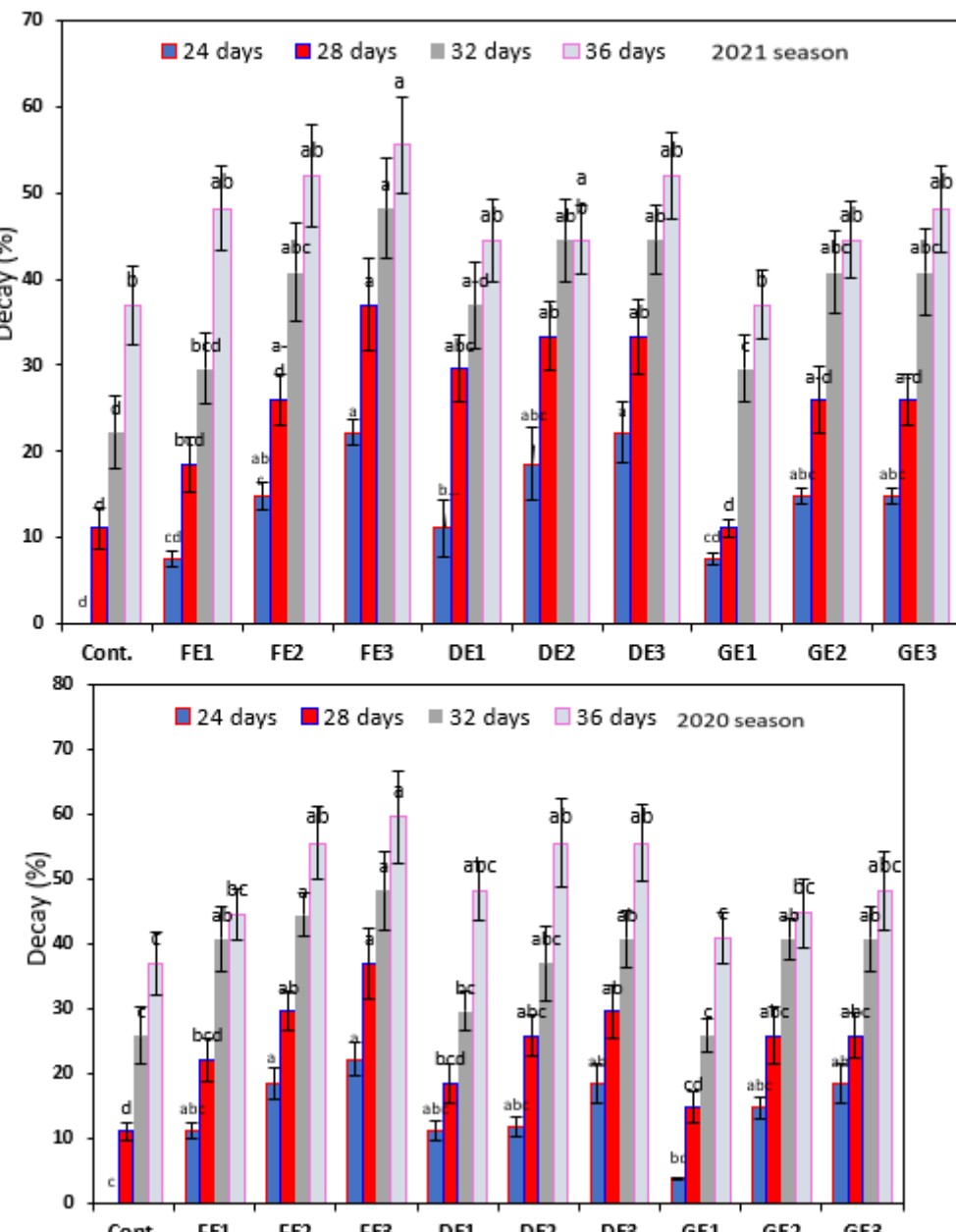

**Figure 5.** Effect of ethrel and ethylene on decay rate (%) of the tomato fruits 8, 16, 24, and 32 days after the treatment in 2020 and 2021. Means followed by the same letter in same storage time are not significantly different at the 0.05 level, according to Duncan's multiple range test. Treatments (in ppm): foliar ethrel included FE1 (2500), FE2 (5000), and FE3 (7500); dipping ethrel included DE1 (1000), DE2 (1500), and DE3 (2000); and ethylene gases were GE1 (100), GE2 (200), and GE3 (300).

### 3.1.5. Fruit Color

Color indexes were evaluated as 10 and 30 after the treatments. All ethrel and ethylene treatments significantly increased the fruit redness compared to the control (Figure 6). This was clearer at day 30 than at day 10. A high redness score reflects a color shift, which indicates a maturation. The highest red color and color index scores were recorded after the ethylene treatment at the highest dose (300 ppm) and after the foliar ethrel application at the highest dose (7500 ppm), followed by ethrel gas at lower doses (1000 and 2000 ppm) and the control. The opposite trend was observed for both the lightness value ("L" value) and the yellowness value ("b" value), which gradually decreased with increasing storage period. The same trend was noticed on the hue angle value ($h^0$ value), which significantly decreased

with increasing doses. It is obvious that the values of "b", "L", and "h⁰" decreased over time and the control showed the highest values. The lowest $h^0$ and 'b' values were obtained after ethylene treatment at the highest dose (GE3).

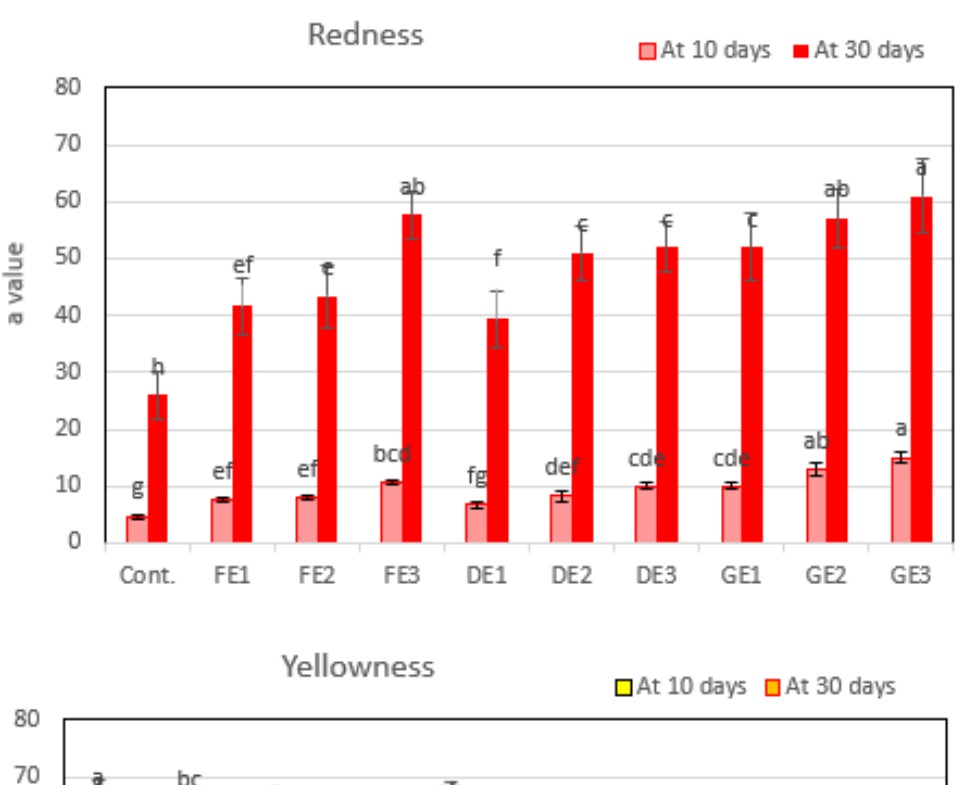

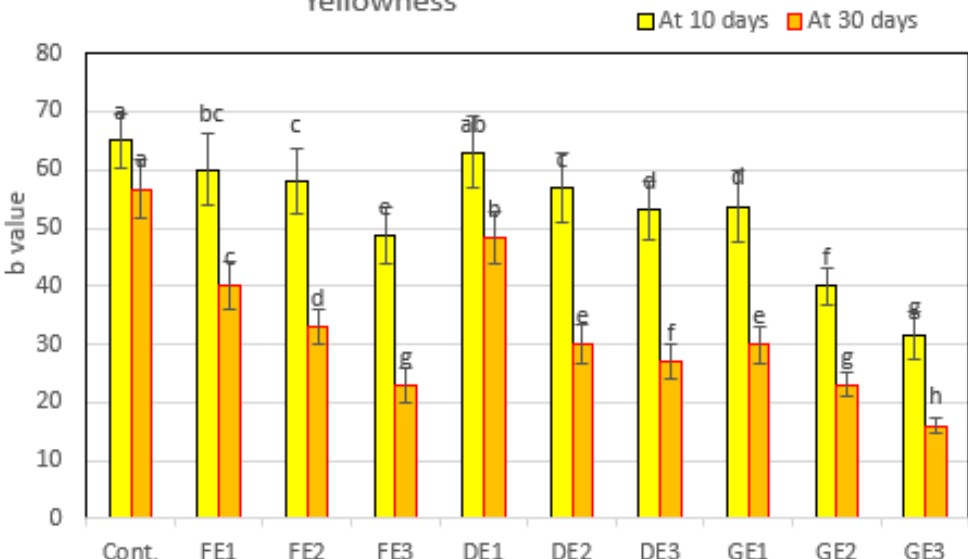

**Figure 6.** *Cont.*

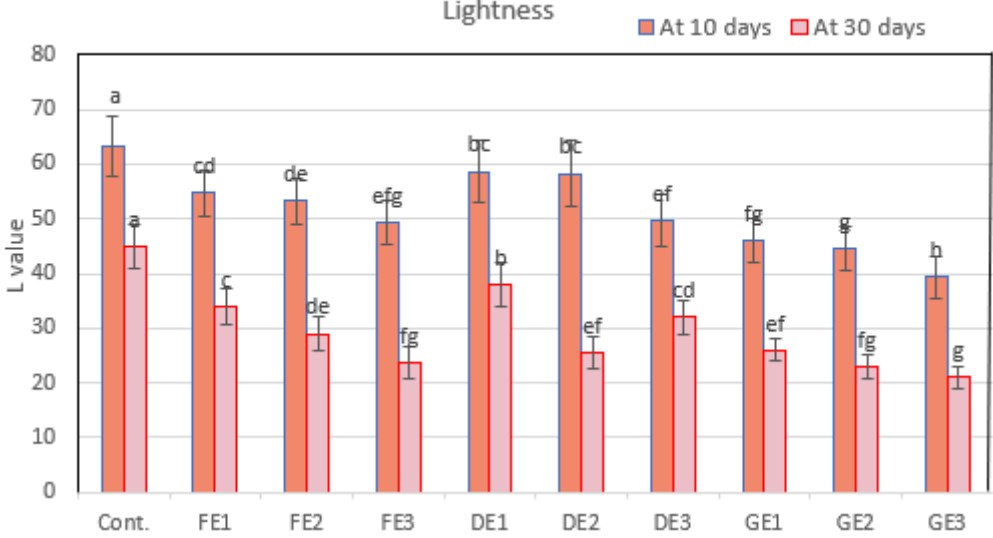

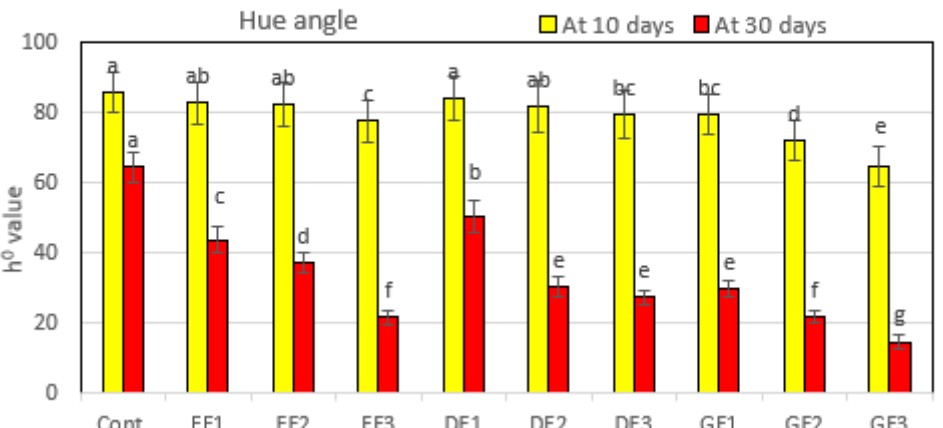

**Figure 6.** Effect of ethrel and ethylene applications on fruit color parameters 10 and 30 days after treatment in the 2021 season. Means followed by the same letter in same storage time are not significantly different at the 0.05 level, according to Duncan's multiple range test. Treatments (in ppm): foliar ethrel included FE1 (2500), FE2 (5000), and FE3 (7500); dipping ethrel included DE1 (1000), DE2 (1500), and DE3 (2000); and ethylene gases were GE1 (100), GE2 (200), and GE3 (300).

### 3.2. Chemical Characterization of Tomato Fruits

#### 3.2.1. Ascorbic Acid Content

In general, data presented in Table 4 show that the ascorbic acid content gradually decreased over time up to day 32 in both seasons. Compared to the control, the applied doses of ethrel and ethylene significantly affected the ascorbic acid content at all sampling dates compared to the control. The highest value was obtained in tomato fruits treated with ethylene gas at the highest dose (GE3) and this result was noticed at all sampling dates. The lowest values were obtained in untreated fruits in both seasons. Ethrel foliar applications and dipping treatments produced intermediate values.

**Table 4.** Effect of ethrel and ethylene applications on ascorbic acid content (mg 100 g$^{-1}$) of the tomato fruits 8, 16, 24, and 32 days after the treatment in 2020 and 2021.

| Treatments | Ascorbic Acid Content (mg 100 g$^{-1}$) of Stored Tomato Fruits after Treatment | | | | | | | |
|---|---|---|---|---|---|---|---|---|
| | 8 Days | 16 Days | 24 Days | 32 Days | 8 Days | 16 Days | 24 Days | 32 Days |
| | | 2020 Season | | | | 2021 Season | | |
| Control | 14.1 e | 13.20 h | 12.97 e | 12.4 d | 17.2 d | 14.0 c | 12.0 c | 11.0 c |
| FE1 | 19.9 b | 28.00 bc | 21.03 b | 18.1 ab | 22.8 ab | 19.0 ab | 16.8 ab | 14.4 a |
| FE2 | 16.1 cd | 18.90 fg | 14.40 de | 13.0 d | 20.0 bc | 17.0 abc | 15.5 abc | 12.0 b |
| FE3 | 15.0 d | 17.60 g | 13.5 e | 13.0 d | 18.0 cd | 16.0 abc | 12.3 c | 11.0 d |
| DE1 | 16.3 cde | 17.60 g | 15.10 d | 12.5 d | 18.0 cd | 16.0 abc | 14.3 abc | 12.0 b |
| DE2 | 17 bcde | 21.10 ef | 15.80 d | 14.7 bcd | 20.0 bc | 17.0 abc | 14.4 abc | 12.5 b |
| DE3 | 18.1 bcd | 23.80 de | 17.60 c | 16.4 bc | 20.0 bc | 17.0 abc | 15.5 abc | 13.0 ab |
| GE1 | 18.1 bcd | 22.80 e | 17.60 c | 15.8 cd | 20.0 bc | 17.0 abc | 15.9 abc | 13.0 ab |
| GE2 | 22.7 a | 30.80 b | 24.00 a | 19.1 a | 25.6 a | 20.0 a | 17.8 a | 14.2 a |
| GE3 | 24.0 a | 35.20 a | 24.00 a | 19.5 a | 26.0 a | 20.0 a | 18.0 a | 14.4 a |
| F-test | ** | ** | ** | ** | ** | ** | ** | * |

Treatments (in ppm): foliar ethrel included FE1 (2500), FE2 (5000), and FE3 (7500); dipping ethrel included DE1 (1000), DE2 (1500), and DE3 (2000); and ethylene gases were GE1 (100), GE2 (200), and GE3 (300). Symbols * and ** means significant at $p < 0.05$ and, $p < 0.01$, respectively. Means followed by the same letter in same column are not significantly different at the 0.05 level, according to Duncan's multiple range test.

### 3.2.2. Total Soluble Solids (TSS)

The same trend that was described for ascorbic acid was observed for the TSS. TSS values increased by increasing doses of ethrel and ethylene (Table 5). The highest values were noticed after ethylene gas application at the highest dose (GE3). The untreated fruits recorded the lowest values.

**Table 5.** Effect of ethrel and ethylene applications on TSS (%) of the tomato fruits 8, 16, 24, and 32 days after the treatment in 2020 and 2021.

| Treatments | TSS Content (%) of Stored Tomato Fruits after Treatment | | | | | | | |
|---|---|---|---|---|---|---|---|---|
| | 8 Days | 16 Days | 24 Days | 32 Days | 8 Days | 16 Days | 24 Days | 32 Days |
| | | 2020 Season | | | | 2021 Season | | |
| Control | 4.5 c | 3.97 d | 3.9 ab | 3.60 b | 4.77 e | 5.0 f | 4.27 e | 4.07 e |
| FE1 | 5.00 ab | 5.07 a | 4.50 ab | 4.00 ab | 5.83 bcd | 6.1 b–f | 5.33 c | 5.50 ab |
| FE2 | 4.90 abc | 4.20 bc | 3.93 ab | 3.70 b | 5.00 e | 5.83 c–f | 5.10 cd | 4.50 de |
| FE3 | 4.50 c | 4.10 bcd | 3.87 b | 3.67 b | 5.10 e | 5.67 ef | 4.60 de | 4.20 e |
| DE1 | 4.80 abc | 4.67 abcd | 4.07 ab | 3.83 ab | 5.77 cd | 6.6 a–e | 5.00 cd | 4.57 de |
| DE2 | 4.83 abc | 4.77 abc | 4.30 ab | 3.90 ab | 6.00 bc | 6.8 a–e | 5.93 b | 4.70 cde |
| DE3 | 4.73 abc | 4.70 abcd | 4.33 ab | 4.03 ab | 5.10 e | 5.9 b–f | 5.10 cd | 4.40 de |
| GE1 | 4.93abc | 4.80 abc | 4.27 ab | 4.17 ab | 6.00 bc | 6.67 a–e | 6.00 b | 5.00 bcd |
| GE2 | 5.07 a | 5.07 a | 4.40 ab | 4.20 ab | 6.40 ab | 7.0 ab | 6.00 b | 5.70 a |
| GE3 | 5.10 a | 5.27 a | 4.80 a | 4.47 a | 6.83 a | 7.27 a | 6.83 a | 5.80 a |
| F-test | * | ** | * | * | ** | ** | ** | * |

Treatments (in ppm): foliar ethrel included FE1 (2500), FE2 (5000), and FE3 (7500); dipping ethrel included DE1 (1000), DE2 (1500), and DE3 (2000); and ethylene gases were GE1 (100), GE2 (200), and GE3 (300). Symbols * and ** means significant at $p < 0.05$ and, $p < 0.01$, respectively. Means followed by the same letter in same column are not significantly different at the 0.05 level according to Duncan's multiple range test.

### 3.2.3. Total Titratable Acidity (TTA)

Overall, the acidity values (TTA) decreased over time, and the lowest values were obtained 32 days after treatments. So, the acidity values decreased with the fruit ripening. Interestingly, the main trend of acidity of tomato fruits was not the same as for the previous values of ascorbic acid and TSS content (Table 6). During entire storage periods, the highest acidity value was observed in untreated fruits (control) followed by fruits dipped in ethrel at applied doses of 1000 and 1500 ppm. Both of the ethrel treatments (foliar and gas applications) resulted in the lowest acidity values during all storage periods in both seasons.

**Table 6.** Effect of applied ethrel and ethylene doses on acidity (%) of the tomato fruits 8, 16, 24, and 32 days after the treatment in 2020 and 2021.

| Treatments | Acidity Content (%) of Stored Tomato Fruits after Treatment | | | | | | | |
| | 8 Days | 16 Days | 24 Days | 32 Days | 8 Days | 16 Days | 24 Days | 32 Days |
| | | 2020 Season | | | | 2021 Season | | |
| Control | 1.090 a | 0.929 a | 0.832 a | 0.768 a | 1.470 a | 1.280 a | 0.960 a | 0.896 a |
| FE1 | 0.960 e | 0.832 b | 0.768 b | 0.640 b | 1.090 c | 0.960 b | 0.640 d | 0.576 ef |
| FE2 | 0.897 f | 0.768 c | 0.640 d | 0.619 c | 1.030 c | 0.832 bc | 0.640 d | 0.512 f |
| FE3 | 0.832 g | 0.640 e | 0.640 d | 0.576 d | 0.832 d | 0.704 c | 0.640 d | 0.512 f |
| DE1 | 1.090 b | 0.832 b | 0.768 b | 0.640 b | 1.280 b | 0.960 b | 0.896 ab | 0.832 ab |
| DE2 | 1.020 c | 0.832 b | 0.661 c | 0.640 b | 1.280 b | 0.960 b | 0.768 c | 0.704 cd |
| DE3 | 1.000 d | 0.768 c | 0.640 d | 0.640 b | 1.150 bc | 0.896 b | 0.768 c | 0.640 de |
| GE1 | 0.960 e | 0.725 d | 0.640 d | 0.576 d | 1.280 b | 1.130 a | 0.960 a | 0.875 ab |
| GE2 | 0.896 f | 0.640 e | 0.576 e | 0.448 e | 1.090 c | 0.960 b | 0.832 bc | 0.768 bc |
| GE3 | 0.832 g | 0.640 e | 0.512 f | 0.384 f | 1.030 c | 0960 b | 0.832 bc | 0.768 bc |
| F-test | ** | ** | ** | ** | ** | ** | ** | ** |

Treatments (in ppm): foliar ethrel included FE1 (2500), FE2 (5000), and FE3 (7500); dipping ethrel included DE1 (1000), DE2 (1500), and DE3 (2000); and ethylene gases were GE1 (100), GE2 (200), and GE3 (300). Symbol ** means significant at $p < 0.01$. Means followed by the same letter in same column are not significantly different at the 0.05 level, according to Duncan's multiple range test.

### 3.2.4. Fruit Pigments

Some tomato fruit pigments (i.e., total chlorophyll, carotene and lycopene) were measured after 28 days from starting the storage period in 2021 when the differences among the treatments were obvious (Figure 7). Generally, all ethrel- and ethylene-applied doses caused a significant reduction in the chlorophyll content of fruits compared with untreated ones (the control). Moreover, the statistical analysis of the present data indicated that ethylene gas treatments caused a higher reduction in chlorophyll than those other ethylene treatments. On the other hand, all higher levels from each applied dose or treatment (i.e., ethrel and ethylene) caused the highest effect in decreasing chlorophyll content. In the same way, carotene content has the same trend as chlorophyll, where it was also significantly decreased by increasing applied doses of both ethrel and ethylene (Figure 7). Ethylene gas applications, especially applied 200 and 300 ppm, resulted in the biggest reduction in carotene content, while untreated fruits showed the highest carotene value without significant differences with ethrel foliar treatment (2500 ppm) and dipping treatment (1000 ppm). On the contrary, lycopene content was significantly increased with all ethrel applications compared with control fruits, which recorded the lowest value of carotene. The highest lycopene content was achieved by ethylene gas application (300 ppm) in comparison with other ethrel applications.

### 3.2.5. Enzymes Activity

The relationship between ethrel- and ethylene treatments and the selected enzymes activity measurements are shown in Figure 8. Both the polygalacturonase (PG) and pectin methyl esterase (PME) activities were measured as an important model for the enzyme activity under the storage conditions of tomato fruits under application of ethrel and ethylene treatments. All measured values of PG after 21 days from storing fruits were higher than those obtained after 10 days of treated tomato fruits regardless of the treatment, whereas the opposite trend was observed for the PME activity. For PG activity, the highest enzyme value (30 and 35 ppm) was obtained by applying FE3 (7500 ppm) to tomato fruits after 10 and 21 days from storing, respectively. The same trend was observed for PME activity, where the highest values (26 and 23 ppm) were also recorded for FE3 (7500 ppm) in tomato fruits after 10 and 21 days from storing, respectively. The activity of PME was decreased by increasing ripening rate from 10 to 21 days during the storage period; however, PG activity was taking an opposite trend and showed an increase with the ripening stage. Ethylene gas treatments (GE1, GE2, and GE3) chronicled low values of both

enzymes without significant differences with control fruits (untreated fruits). However, under ethylene treatments or applied doses from GE1 to GE3, the enzyme activity of both PG and PME values were increased by increasing the applied dose of ethylene from 100 to 300 ppm in the two studied storage periods (10 and 21 days).

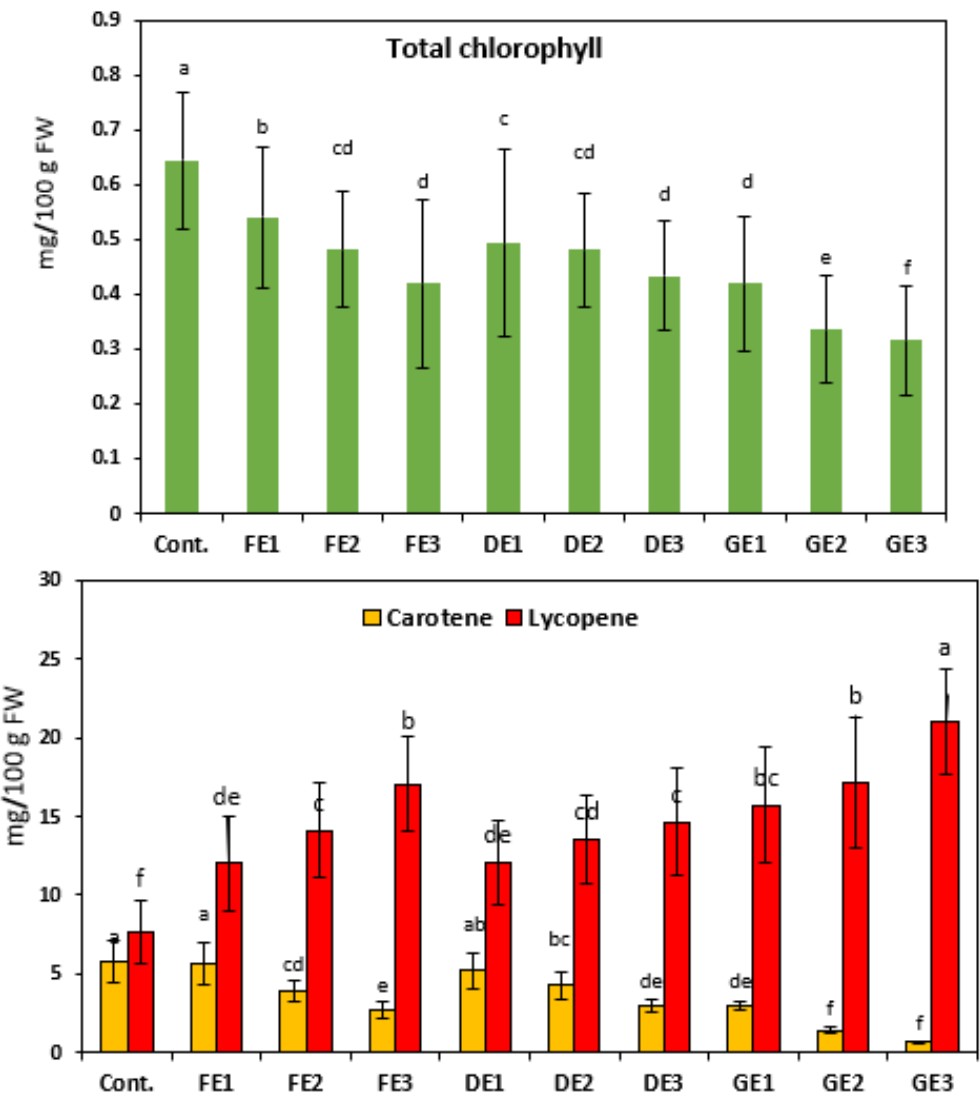

**Figure 7.** Effect of ethrel applications on total chlorophyll and carotene contents (mg 100 g$^{-1}$ FW) of the tomato fruits 28 days after the treatments in the 2021 season. Means followed by the same letter in same storage time are not significantly different at the 0.05 level, according to Duncan's multiple range test. Treatments (in ppm): foliar ethrel included FE1 (2500), FE2 (5000), and FE3 (7500); dipping ethrel included DE1 (1000), DE2 (1500), and DE3 (2000); and ethylene gases were GE1 (100), GE2 (200), and GE3 (300).

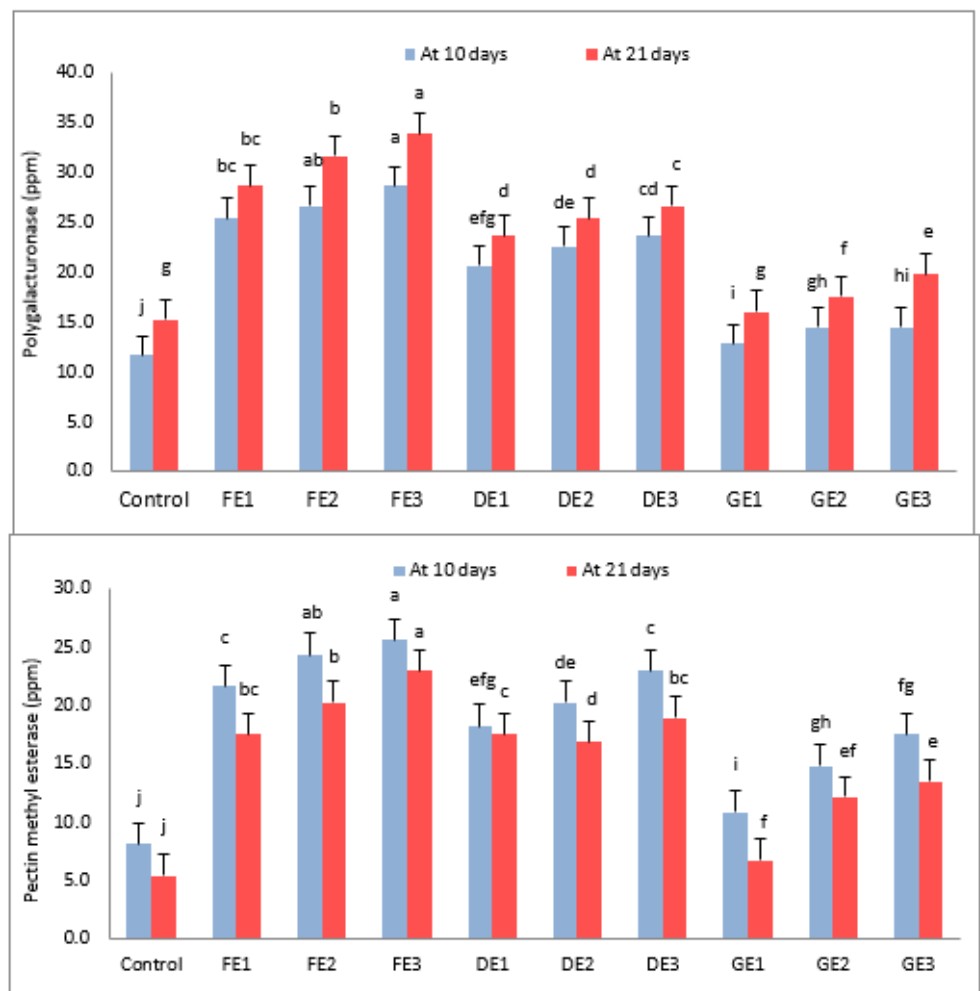

**Figure 8.** Impact of different applied doses of ethrel and ethylene treatments on the activity of polygalacturonase and pectin methyl esterase 10 and 21 days after the treatments in the 2021 season. Means designed by the same letter (at the same storage time) are not significantly different at the 0.05 level, according to Duncan's multiple range test. Treatments (in ppm): foliar ethrel included FE1 (2500), FE2 (5000), and FE3 (7500); dipping ethrel included DE1 (1000), DE2 (1500), and DE3 (2000); and ethylene gases were GE1 (100), GE2 (200), and GE3 (300).

## 4. Discussion

### 4.1. Impact of Ethylene on Ripening of Cold Stress Grown Tomato Fruits

We will now discuss which method was effective in promoting the physical and chemical properties involved in tomato fruits ripening under cold stress. The selected period represents the tomato production during the early summer season in Egypt, which included mean monthly air temperature from January to May in the range from 12.5 to 24.5 °C. In January, the mean daily temperature ranged from 7 to 21 °C. To speed up the ripening, farmers are looking for good and safe methods. The timing of tomato ripening is very important for fruits quality. Tomato, as a climacteric fruit, should be well understood when it comes to its ripening behavior. Fruits can be harvested before full maturity and then ripen on its way to the consumers. It is useful to find suitable ways to regulate this ripening and at the same time maintain the quality [22]. Our study showed that most physical fruit ripening parameters were accelerated by applying ethylene gas after harvesting. This treatment was better than foliar spraying and dipping in ethrel solutions. The ripening was fastest for fruits exposed to the highest ethylene gas doses (GE3; 300 ppm) (Table 2). This result is in agreement with the study of Dhall and Singh [23], who reported that ethylene is better than ethrel in ripening tomato fruits. They reported that only 9 days were

needed for adequate fruit ripening with the use of ethylene gas, which corresponds well with our 8 days. The period for exposure to ethylene gas in our study was 24 h, which was enough for this plant hormone, as reported by Aprianti and Bintoro [7]. Studying the ripening at room temperature (20–30 °C), as we did, is suitable for ripening process transformations [24]. The ripening process of tomato fruits is linked to the climacteric properties of these fruits, which is characterized by having a high respiration rate and a production of ethylene [25].

### 4.2. Tomato Fruit Quality Influenced by Methods of Ethylene Treatment

Over storage time, which in our study was up to 32 days, there was an increase in weight loss (Figure 3). The lowest weigh loss was recorded after exposure to ethylene gas (GE1; 100 ppm). The weight losses increased by increasing the doses in each of the methods used for producing ethylene (Figure 3). This may be due to the increased respiration rates associated with the ripening process [23]. Concerning the fruit firmness, which is an important indicator of fruit freshness, it decreased over time in all treatments (Table 3). The highest values were observed after the addition of ethylene gas at the lowest dose (GE1; 100 ppm). This may be due to a conversion of insoluble proto-pectin into soluble pectin, losing pectin in the middle lamella of the cell wall, and thereby reducing the cell wall integrity and softening the fruits [23]. A similar result was observed in the firmness of tomato fruits by Ali et al. [5]. Tomato fruit decay was the fastest after foliar spraying of ethrel (FE3; 2000 ppm) (Figure 4). Both the control and ethylene gas resulted in slower decay rates than ethrel. The main reason for fruit decay is microorganisms, including fungi (*Alternaria*, *Botrytis*, *Geotrichum*, and *Rhizopus*) and bacteria [26].

Regarding tomato fruit color, such a parameter is the most important quality attribute of tomato fruits. Many parameters could be found in the color index, including redness, yellowness, lightness, and the hue value (Figure 5). Over storage time, the redness increased and the redness was the highest after applying ethylene gas (GE1; 100 ppm), whereas the opposite trend was observed in case of yellowness, lightness, and the hue, which recorded the lowest values after 30 days for GE3 (300 ppm). The main source of fruit color is the pigments and their changes during maturation. These pigments may include chlorophylls, carotenoids, β-carotene, and lycopene, which may change during fruit storage [27].

The accumulation of both chlorophyll and carotene decreased by increased doses of ethylene and ethrel, whereas the opposite trend was observed to lycopene (Figure 6). The fruit ripening is stimulated by the ethylene production, which accelerates the change in fruit color (pigments) from green to red by a degradation of the pigments, mainly chlorophyll and carotene through chlorophyllase enzyme [23], or modulating the balance of auxin-ethylene [28]. Therefore, the tomato fruit quality during postharvest storage may have certain changes in fruit pigmentation (mainly skin color), which is associated with the ripening stage, contents of phenolic compounds, lycopene, and antioxidant activity [29]. The obtained results in our study are in accordance to other studies, such as Park et al. [29].

Concerning the biochemical attributes of the tomato fruits, they include ascorbic acid, acidity, TSS, fruit pigments, and enzymes. Concerning ascorbic acid, TSS, and acidity (organic acids), which are all important taste characteristics of tomato fruits, a decrease over storage time is expected [27]. In our study, the values of these parameters decreased over storage time from 8 to 32 days after harvesting (Tables 4–6). The highest values were recorded by ethylene gas (GE3; 300 ppm) in the case of ascorbic acid (vitamin C) and TSS, whereas they were recorded by the control in the case of acidity. The observed decrease in tomato fruit acidity was gradual during the ripening period, which could be attributed to the utilization of organic acids in the reaction of pyruvate decarboxylation during the fruit-ripening process. Under faster respiration rates, more acids could be stored in cell vacuoles due to increased membrane permeability [23]. During tomato fruit ripening, an increase in TSS could be attributed to loss of water and hydrolysis of polysaccharides and starch into soluble sugars, whereas a decrease in TSS could be due to fruit respiration [23]. Regarding the ascorbic acid (vitamin C) over storage time, the values decreased over storage time due

to the breakdown of this vitamin. These results are in agreement with the results obtained by Dhall and Singh [23] and Ali et al. [5].

During the ripening process, several biochemical reactions take place and changes in enzyme activities could happen, such as the activity of both polygalacturonase (PG) and pectin methyl esterase (PME). These enzyme activities may control the ripening process of tomato fruits. In the present study, the activity of PG increased by storage time from 10 to 21 days in all applied doses under the different ethylene-producing methods, whereas the opposite trend was reported for PME enzyme activity (Figure 7). It was also noticed that the method, which applied ethylene as gas, recorded the lowest values in both enzymes, whereas the foliar application of ethrel resulted in the highest values. The key for tomato fruits ripening mainly depended on the activation and/or inhibition of certain enzymes, which can accelerate or slow down the ripening process. All previous physical, chemical, and biochemical parameters attributed to the tomato fruits' ripening need a certain enzyme, which again is controlled by specific plant genes. These bio-reactions in tomato fruits are influenced by environmental conditions or storage conditions, such as temperature and light, beside the fruits' physiological activities. For example, a high storage temperature may accelerate the fruit ripening by increasing the activity of many enzymes, such as polygalacturonase and pectin methyl esterase [30,31].

### 4.3. Ethylene for Industrial Ripening

To what extent is ethylene common to use for industrial ripening? Many vegetable and fruit crops have been successfully ripened using ethylene. In Egypt, ethylene has been applied for a long time for fruits, such as banana and persimmons, and also for some other fruits and vegetables. This artificial ripening has been used as an economic approach as it only takes 3–5 days from the ethylene treatment of banana and persimmons to ripen fruits to being sold, whereas it takes from 15 to 30 days without ethylene, and the ripening can still be incomplete. Ethylene that is produced from the applied methods in our study can regulate the ripening of tomato fruits, and a balance between the action of this plant hormone and other phytohormones (mainly auxins) will regulate the ripening through the activity of the involved enzymes [32]. Therefore, banana and persimmons must be harvested at morphological maturity by exposure to ethylene, which can save time and effort, as well as high net profit. Depending on the crop, the used rate of ethylene may reach 1000–1500 ppm, which differs according to the crop-growing period as well.

### 4.4. Artificial Ripening of Tomato Fruits

The cost of this artificial repining treatment can be covered by the high net profit gained due to the early production of tomato. Therefore, the production of tomato in open fields, as found in northern Egypt during the early summer season, could be harvested at green mature stage and the ripening can be accelerated by using ethylene. A search on Clarivate analytics, which houses more than 12,000 international journals, showed an increase in interest for the topic of our paper. We used the advanced search function and the script "TS = (Tomato AND ethylene AND ripen*)" and refined the search by articles only and the categories agronomy, horticulture, or food science technology only, which resulted in 535 publications in total. Although there is a huge number of publications, there is a knowledge gap on practical implementations, as the use of doses or formula. This study may also agree with the results of Shu et al. [33], who reported about the role of ethylene in enhancing tomato fruit tolerance to low-temperature stress. They confirmed that lower temperatures can inhibit increasing ethylene content in fruits by regulating the antioxidant enzymes. Therefore, applying exogenous ethylene can enhance the ripening of tomato fruits, which were grown under cold stress by reducing the damage resulted from cold stress.

## 5. Conclusions

In the current study, the main target was to accelerate the ripening using three different methods that are producing ethylene (i.e., foliar application of ethrel on cultivated plants in the field, dipping harvested tomato fruits in ethrel solution, and applying ethylene gas to harvested fruits in storage). The ethylene gas method recorded the overall best results for fast ripening compared to the other methods and the control. We explain the result as a direct impact of the ethylene gas on the ripening process through enhancing different physical, chemical, and biochemical properties, whereas the production of ethylene from ethrel needs more time. Further studies are needed to address other amendments that can be applied to promote the ripening of the tomato fruits, for example, if new approaches, such as nanomaterials, may be used in cultivation and post-harvest treatments of tomato fruits.

**Author Contributions:** Conceptualization and visualization, Y.B. and S.Ø.S.; methodology, S.O.; software, E.-S.E.-S.; validation, A.E., E.-S.E.-S., S.Ø.S. and H.E.-R.; formal analysis, S.O. and Y.B.; investigation, A.E.; resources, H.E.-R.; data curation, A.E.; writing—original draft preparation, H.E.-R.; writing—review and editing, all authors.; visualization, S.Ø.S.; supervision, A.E. and Y.B.; project administration, S.Ø.S.; funding acquisition by S.Ø.S. All authors have read and agreed to the published version of the manuscript.

**Funding:** There is no external funding.

**Institutional Review Board Statement:** Not applicable.

**Data Availability Statement:** Not applicable.

**Acknowledgments:** Authors acknowledge technical and administrative support from our institutions. The authors thank the staff members of Physiology and Breeding of Horticultural Crops Laboratory, Department. of Horticulture, Faculty of Agriculture, Kafrelsheikh University, Kafr El-Sheikh, Egypt for conducting the biochemical assays and other parameters.

**Conflicts of Interest:** There is no conflict of interest among the authors.

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
