# Peer review of "Regulating Enzymatic Antioxidants, Biochemical and Physiological Properties of Tomato under Cold Stress: A Crucial Role of Ethylene"

_agriculture, doi:10.3390/agriculture13020266_

Round 1

Reviewer 1 Report

Drastic improvement in the English language is needed, please get it revised throughout the text. I have made several corrections at some places in the beginning of the MS.

Do not write the values (highest or lowest) in the results section as it is already mentioned in the table.

Section 4.1 (line 450-463) are already mentioned in the Introduction, so no need to repeat it here.

The author should carry out ANOVA for each trait taking year also as one of the variables and present the results in one table. If the difference between the years is not significant (which should be the case), then they can take the average of two years to write the results. This will also reduce the number of tables/graphs.

In conclusion, line 596-601, maybe deleted. 

Author Response

Reviewer 1#

Comments and Suggestions for Authors

Drastic improvement in the English language is needed, please get it revised throughout the text. I have made several corrections at some places in the beginning of the MS.

Response: the MS was edited and improved. We had an extensive editing action that involved consultations with a native English-speaking person. We made more than 100 of changes and all changes are marked with the Track Changes function in Word or with yellow highlights in the running text.

The edited version was submitted in the supplementary materials as well!

Do not write the values (highest or lowest) in the results section as it is already mentioned in the table.

Response: thanks for your comment! We have removed the detailed values at several places throughout the result section. The changes are marked with the Track Changes function in Word or with yellow highlights in the running text.

Section 4.1 (line 450-463) are already mentioned in the Introduction, so no need to repeat it here.

Response: thanks for your comment! Deleted, thanks!

The author should carry out ANOVA for each trait taking year also as one of the variables and present the results in one table. If the difference between the years is not significant (which should be the case), then they can take the average of two years to write the results. This will also reduce the number of tables/graphs.

Response: thanks for your comment!

We analyzed the data statistically with considering the season as a factor! We found a significant difference in some parameters, which will increase the tables because we should add the table of interaction as well like the table below. So, we could not apply this statistical analysis!

Table 2. Effect of ethrel and ethylene applications on the ripening % of tomato fruits during the storage period in the 2020 and 2021 seasons

Factors

Ripening (%) of stored tomato fruits after days

4 days

8 days

12 days

Years

1-      2020 season

2-      2021 season

F. test

33.3 b

39.1 a

**

63.0 a

58.2 b

*

100.0

100.0

NS

Treatments

1-      Control

2-      FE1

3-      FE2

4-      FE3

5-      DE1

6-      DE2

7-      DE3

8-      GE1

9-      GE2

10- GE3

F. test

0.00 f

16.7 e

22.2 e

48.2 bc

22.2 e

37.0 d

44.1 c

38.9 d

50.1 b

63.0 a

**

16.7 h

32.5 g

51.9 e

87.1 bc

38.8 fg

51.9 e

66.7 de

70.4 cd

90.8 ab

100.0 a

**

100.0

100.0

100.0

100.0

100.0

100.0

100.0

100.0

100.0

100.0

NS

Interactions

2020

1-      Control

2-      FE1

3-      FE2

4-      FE3

5-      DE1

6-      DE2

7-      DE3

8-      GE1

9-      GE2

10-  GE3

0.00

14.8

18.5

48.2

18.5

37.0

40.0

40.8

51.9

63.0

0.00

18.5

25.9

48.2

25.9

37.0

48.2

37.0

48.2

63.0

NS

18.5 g

37.0 f

55.6 d

92.6 ab

37.0 f

51.9 de

77.8 c

70.4 c

88.9 b

100.0 a

14.8 g

25.9 g

48.2 e

81.5 bc

40.5 ef

51.9 de

55.6 d

70.4 c

92.6 ab

100.0 a

*

100.0

100.0

100.0

100.0

100.0

100.0

100.0

100.0

100.0

100.0

100.0

100.0

100.0

100.0

100.0

100.0

100.0

100.0

100.0

100.0

NS

2021

1-      Control

2-      FE1

3-      FE2

4-      FE3

5-      DE1

6-      DE2

7-      DE3

8-      GE1

9-      GE2

10-  GE3

F. test

Treatments (in ppm): foliar ethrel included FE1 (2500), FE2 (5000), and FE3 (7500), dipping ethrel included DE1 (1000), DE2 (1500), and DE3 (2000), whereas ethylene gas were GE1 (100), GE2 (200), GE3 (300).

NS means non-significant, whereas ** means significant at p < 0.01

Means followed by the same letter in same column are not significantly different at the 0.05 level according to Duncan's multiple range test.

The complete file of statistical analyses based on two seasons, was attached in the supplementary material!

In conclusion, line 596-601, maybe deleted. 

Response: thanks for your comment! Deleted, thanks!

Comments in the pdf file:

Lines 17-18, 22, 23, 24, 25-33.

Response: thanks for your comment! Corrected and added, thanks!

Lines 30-33:

This cannot be the conclusion as there are thousands reports as well as ethylene is commercially being used as ripening hormone for fruits including tomato.

Response: thanks for your comment! Corrected, thanks!

Lines 42: "poor man’s orange". Or apple

Response: thanks for your comment! According to the ref. is orange, thanks!

Lines 44-45:

The benefits of tomato can be said in two sentences, some of these are repeated. make it crisp.

Response: thanks for your comment!

This part was re-written again to avoid repetition and to be fit, thanks!

These fruits have many benefits for human health because they are rich in vitamins (e.g., A and C), and antioxidants (e.g., lycopene, and β-carotene). Thus, these attributes can decrease the risk of chronic diseases (e.g., cancer and heart disease), prevent and deactivate free radicals or reactive oxygen species, and act as an effective eliminator of superoxide, hydrogen peroxide, singlet oxygen and other free radicals [4,5]. Therefore, it is recommended daily to consume about 100 g of tomato fruits to improve the human immune system, lower cholesterol and reduce pressure blood [6].

Lines 90 – 98:

This para needs to be re-written with clear objectives. The author should discuss the various methods of ethylene application from the earlier studies, compare and then put forward their objective to fill the research gap. It is well known that ethylene is used as fruit ripening hormone including tomato fruits, nothing new. So, find out the research gaps.

Response: thanks for your comment!

This section was re-written again to meet the main objectives of this study as follows:

Several studies are available on the effect of ethylene on fruit ripening, but our study will discuss three different application methods and different applied concentrations under cold stress, as a main target. To achieve this main aim, various physical and chemical and enzymatic characters were investigated as response variables. The current study will also focus on the role of ethylene in accelerating tomato fruit ripening and improving quality under cold stress (i.e., early summer season) as a Mediterranean climate in Northern Egypt.

Line 126

what these FE, GE and DE stands for?

Response: thanks for your comment!

These are abbreviation or code of the used methods in our work. The meaning of each one was added as follows:

Foliar ethrel (FE), dipping in ethrel (DE) and ethylene gas (GE)

Figure 3: The Y2-axis of 2021 were

Response: thanks for your comment! Each figure has 2 Y axis, one for the temperature and the other for relative humidity, thanks!

Line 167: Days are not 'measured' but 'counted'

Response: thanks for your comment! Corrected, thanks!

Line 170: which company/make? add a little of description how the firmness was measured.

Response: thanks for your comment!

The firmness of tomato fruits was measured using Push-pull Dynamometer (Models FD, and FDN, Italy). The dynamometer is calibrated for use in Horizontal position Reading was recorded after pushing and pulling into the fruits.

Thanks so much for all these valuable comments. We hope that the new version is acceptable for publication for a broader audience and further critical comments. We are fully aware of its limitations but still believe our results are useful for the production of tomato.

Reviewer 2 Report

The authors compared three method of tomato ripening, and found applying ethylene gas is the most effective. In fact, applying ethylene has been widely used for the ripening of fruits. Thus, this work appears to be not novel. Thus, I have no comment on the scientific significance of this work. However, the authors detected a set of physiological indices, which will provide some evidence for us to know the effect of chemical ripening on the quality of fruits.  

I have some concerns:

1. The manuscript does not provide the sample size, so I don't know whether ANOVA, especially LSD posthoc comparison, is suitable.

2. The data lacked standard deviation in some figures.

3. The figures need to be largely improved. The figures in this version is too large and rough. I strongly recommend to improve the quality of figures.

4. Please uniform the style of tables. For example, there has a vertical line in Table 6, but it is not necessary.

5. The mehod of application is not cearly stated.

Author Response

Reviewer 2#

Comments and Suggestions for Authors

The authors compared three method of tomato ripening, and found applying ethylene gas is the most effective. In fact, applying ethylene has been widely used for the ripening of fruits. Thus, this work appears to be not novel. Thus, I have no comment on the scientific significance of this work. However, the authors detected a set of physiological indices, which will provide some evidence for us to know the effect of chemical ripening on the quality of fruits.

Response: thanks for your comment! Corrected, thanks!

I have some concerns:

  1. The manuscript does not provide the sample size, so I don't know whether ANOVA, especially LSD posthoc comparison, is suitable.

Response: thanks for your comment! Corrected, thanks!

  1. The data lacked standard deviation in some figures.

Response: thanks for your comment! completed, thanks!

  1. The figures need to be largely improved. The figures in this version is too large and rough. I strongly recommend to improve the quality of figures.

Response: thanks for your comment! Corrected, and every figure will be fit after formatting of the journal, this is tentative form, thanks!

We tried to make the figures smaller, but we found that the figures were became non-readable because the values and titles were confused! Please considered this situation, thanks!

  1. Please uniform the style of tables. For example, there has a vertical line in Table 6, but it is not necessary.

Response: thanks for your comment! Corrected, thanks!

  1. The method of application is not clearly stated.

Response: thanks for your comment! Corrected, thanks!

All details were explained with some changes including the place in which this work was carried out to be fit in the revised MS in lines from 114 – 140

And in Figure 1. The description of the applied treatments and different applied doses (ppm) were put!

Thanks so much for all these valuable comments. We hope that the new version is acceptable for publication for a broader audience and further critical comments. We are fully aware of its limitations but still believe our results are useful for the production of tomato.

Round 2

Reviewer 1 Report

There is a significant improvement in the revised MS. However, there are a few minor corrections which can be done before accepting the MS.

1. In figures, the Y-axis titles are to be properly spaced as it is overlapping with the number scale or with another graph. Check for all figures.

2. The sub-headings in Discussion can be clubbed under titles as 'impact of ethylene on ripening of cold stress grown tomato fruits' 

4.2, 4.3, 4.4, 4.5 and 4.6 can be clubbed under sub-head 'tomato fruit quality influenced by methods of ethylene treatment'

I did not understand the significance of adding sub-head 4.7 and 4.8. The authors have not used ethylene for industrial ripening so why this subhead, secondly, they can suggest the method of ethylene application for industrial use which can be added as one prospect in the conclusion.

In 4.8, they tried to exhibit the research gap but for this, the suitable place is Introduction not Discussion. 

Providing ANOVA tables in supplementary has no meaning, it can be deleted.

Author Response

Reviewer 1#

Comments and Suggestions for Authors

There is a significant improvement in the revised MS. However, there are a few minor corrections which can be done before accepting the MS.

Response: many thanks!

  1. In figures, the Y-axis titles are to be properly spaced as it is overlapping with the number scale or with another graph. Check for all figures.

Response: many thanks! All figures were checked, thanks again! We made some changes to be clear, thank!

  1. The sub-headings in Discussion can be clubbed under titles as 'impact of ethylene on ripening of cold stress grown tomato fruits'

Response: many thanks! Done!

4.2, 4.3, 4.4, 4.5 and 4.6 can be clubbed under sub-head 'tomato fruit quality influenced by methods of ethylene treatment'

Response: many thanks! Done!

I did not understand the significance of adding sub-head 4.7 and 4.8. The authors have not used ethylene for industrial ripening so why this subhead, secondly, they can suggest the method of ethylene application for industrial use which can be added as one prospect in the conclusion.

In 4.8, they tried to exhibit the research gap but for this, the suitable place is Introduction not Discussion.

Response: many thanks! Please let these sub-sections, because we added based on comments from another reviewer, thanks a lot for you understanding!

Providing ANOVA tables in supplementary has no meaning, it can be deleted.

Response: many thanks! Done we will remove them, thanks

Thanks so much for all these valuable comments. We hope that the new version is acceptable for publication for a broader audience and further critical comments. We are fully aware of its limitations but still believe our results are useful for the production of tomato.

Reviewer 2 Report

The authors made a response to my comments. I have no further concern.

Author Response

Reviewer 2#

Comments and Suggestions for Authors

The authors made a response to my comments. I have no further concern.

Response: Many thanks for your response!
